# A Nuclear Magnetic Resonance Spectroscopy Method in Characterization of Blood Metabolomics for Alzheimer’s Disease

**DOI:** 10.3390/metabo12020181

**Published:** 2022-02-15

**Authors:** JianXiang Weng, Isabella H. Muti, Anya B. Zhong, Pia Kivisäkk, Bradley T. Hyman, Steven E. Arnold, Leo L. Cheng

**Affiliations:** 1Department of Pathology, Massachusetts General Hospital, Harvard Medical School, Boston, MA 02115, USA; jweng5@mgh.harvard.edu (J.W.); imuti@mgh.harvard.edu (I.H.M.); abzhong@mgh.harvard.edu (A.B.Z.); 2Department of Neurology, Massachusetts General Hospital, Harvard Medical School, Boston, MA 02115, USA; pkivisakk@mgh.harvard.edu (P.K.); bhyman@mgh.harvard.edu (B.T.H.); searnold@mgh.harvard.edu (S.E.A.); 3Departments of Radiology and Pathology, Massachusetts General Hospital, Harvard Medical School, Boston, MA 02115, USA

**Keywords:** Alzheimer’s disease, metabolomics, nuclear magnetic resonance spectroscopy, blood plasma

## Abstract

There is currently a crucial need for improved diagnostic techniques and targeted treatment methods for Alzheimer’s disease (AD), a disease which impacts millions of elderly individuals each year. Metabolomic analysis has been proposed as a potential methodology to better investigate and understand the progression of this disease. In this report, we present our AD metabolomics results measured with high resolution magic angle spinning (HRMAS) nuclear magnetic resonance (NMR) on human blood plasma samples obtained from AD and non-AD subjects. Our study centers on developments of AD and non-AD metabolomics differentiating models with procedures of quality assurance (QA) and quality control (QC) through pooled samples. Our findings suggest that analysis of blood plasma samples using HRMAS NMR has the potential to differentiate between diseased and healthy subjects, which has important clinical implications for future improvements in AD diagnosis methodologies.

## 1. Introduction

Throughout the development and progression of Alzheimer’s disease (AD)—from the presence of predisposing genetic factors and environmental stimuli to the degeneration of synapses and neurons—the overall metabolic status, or metabolomics, of the brain and other organs changes in AD patients, shifting from the normal homeostasis of healthy individuals to pathological states. For this reason, discovering and understanding AD-associated metabolomic changes in the brain and other peripheral systems, such as in blood, cerebrospinal fluid, and urine, may assist with an early and definitive diagnosis and, more importantly, contribute to the development of precision treatments.

Currently, the major methodologies used in AD metabolomic investigations have been mass spectrometry (MS) and nuclear magnetic resonance (NMR) spectroscopy, each with its strengths and weaknesses. MS is able to quantify metabolites at much lower concentrations than NMR, but its drawbacks are that it is destructive and less capable of investigating the physical and chemical status of the targeted molecules, such as diffusion and pH. Furthermore, unlike MS, NMR requires less sample preparation and has the potential to be implemented in in vivo observations with widely available MRI scanners. Using NMR and MS, AD metabolism and metabolomics have investigated human biofluids ranging from cerebrospinal fluid [1,2,3,4,5,6], to less-invasive blood [7,8,9,10,11,12,13,14,15,16,17,18,19,20,21,22,23,24,25], to entirely non-invasive saliva and urine [26,27].

The development of high-resolution magic angle spinning (HRMAS) NMR for intact tissue analysis [28] further enhanced the NMR capability in evaluating blood specimens, either in the form of serum or plasma. Compared to the typical aqueous solutions suitable for NMR analysis, in which intermolecular interactions can be neglected due to the long distance between molecules-of-interest, these biofluids contain macro-molecules, such as proteins, that introduce non-negligible inter-molecular interactions and result in limited achievable spectral resolution for metabolite differentiations. Studies of solid-state NMR physics identified these inter-molecular interactions to be dependent on the angle between the NMR magnetic field and the interaction axes, known as the “magic angle.” Magic angle spinning (MAS) was developed to eliminate the confounding effect of these inter-molecular interactions and achieve better spectral resolutions by spinning the sample at the magic angle with mechanical force such that the time-average of these inter-molecular interactions is reduced to zero. With biofluid samples, these confounding factors can be overcome by the HRMAS method, which is also capable of producing high-resolution NMR spectra from blood samples with a volume less than a drop of blood, commonly considered to be 50 μL [29,30].

Here, we wish to present our HRMAS NMR studies of human blood plasma samples obtained from AD and non-AD subjects (See Appendix A for patient demography and clinical data), with emphasis on methodological developments, including procedures of quality assurance (QA) and quality control (QC).

## 2. Results

### 2.1. HRMAS NMR Accuracy Analyzed by QA/QC Pooled Plasma Controls

To test and evaluate our systematic QA/QC, we pooled 16 AD and 14 non-AD blood plasmas to create 6 pooled control samples. Figure 1A presents the comparison between the average spectrum of all 35 individual samples, and the average spectrum of the 6 pooled controls. Statistically significant linear correlations with a tangent close to 1.0 (*t* = 0.99 ± 0.03, with *r*^2^ = 0.93 ± 0.02, and *p* = 3.84 ± 3.82 e^−17^) were observed between the spectral intensities of the 36 identified regions measured from the mean, with standard errors (SE) indicated by the horizontal error bars for the 35 individual samples and each of the pooled controls, as shown in Figure 1B. These statistically significant linear correlations with tangents close to 1.0 suggest the overall systematic QA/QC satisfaction of the data. Upon confirming the quality of the data, we proceeded with evaluating the spectral intensity data measured from individual plasma samples. Of special note, the three regions that displayed large SEs all included the measurable lipids.

### 2.2. Differentiation of AD from Non-AD Based on Plasma Metabolomics

Figure 2A compares the averaged spectra of AD (*n* = 16) and non-AD (*n* = 19) plasma, with 36 identified spectral regions, in terms of their means and standard errors, which are presented in Figure 2B according to their statistically significant level (*p* values) obtained from univariate analysis. Among the 36 regions analyzed, four of them presented *p* values < 0.05.

We further evaluated the measured 36 spectral regions for their potential to differentiate AD clinical parameters, including Global Clinical Dementia Rating (CDR), CDR Sum of Boxes (SoB), and Mini-Mental State Exam (MMSE), and found that eight, six, and three of the 36 total regions could differentiate CDR (ANOVA or Wilcoxon/Kruskal–Wallis tests) or linearly correlate with CDR, SoB or MMSE, respectively, with statistical significances, as exemplified in Figure 3.

The calculated PCA results, which used all 36 spectral regions from 35 individual samples, showed that while PC1 could differentiate CDR (Figure 4A), and PC3 linearly correlated with MMSE (Figure 4B), PC2 and PC3 indicated differentiation between AD and non-AD groups (Figure 4C). Furthermore, PC3 was shown to differentiate between the two groups with statistical significance and present an accuracy of 68% according to the ROC analysis.

The contribution of a considered spectral region towards the calculated principal component depends on its loading coefficient, determined through PCA analysis, as well as the mean and the standard deviation calculated for that region with all the involved individual samples. The resulting overall loading factor for the spectral region is the product of the loading coefficient and the ratio of the mean over the standard deviation. Among the 36 overall loading factors for the 36 regions, we examined the top 50% of positively contributing regions (which correspond with AD cases for PC1, and non-AD cases for PC3) and the bottom 50% of the most negatively contributing regions (non-AD cases for PC1 and AD cases for PC3). We further considered the potential contributing metabolites represented by these regions, as indicated in Appendix A. The remaining potential unidirectionally contributing metabolites towards AD and non-AD differentiation from PC1 and PC3 are presented in Figure 4D, according to the order of *p*-values of univariate comparisons between AD and non-AD for each analyzed spectral region. These values were calculated after excluding metabolites that could contribute to both “positive” and “negative” regions. Of note, for both PC1 and PC3, while AD included almost the same spectral regions (11/12), non-AD suggested different regions for these two PCs.

### 2.3. Potential Major Contributing Metabolites and Pathways in Differentiating AD from Non-AD

Potential major contributing metabolites that contribute either to the top 50% or bottom 50% of the major regions shown in Figure 4D are summarized in Figure 5 according to their potential involvement in a number of metabolic pathways, including glycolysis, anaerobic glycolysis, the Krebs cycle, and the urea cycle. The positively contributing metabolites represent the increased abundance observed with AD cases, while negative contributions indicate less abundant metabolites. Examination of the metabolite distributions shown in Figure 5 revealed a few commonalities: metabolites with increased abundance in AD include sugar phosphates and derivatives of glucose from the glycolysis pathway, as well as methylated amino acids; while organic acids in the Krebs cycle were less abundant (see Appendix A for details).

## 3. Discussion

A recognized need exists for better understanding of AD metabolomics that may assist both disease diagnosis and inform the development of improved therapies. In response to this need, blood samples have been identified as the ideal candidate for study, as drawing blood is minimally invasive and easily repeatable. Numerous NMR studies of human blood serum samples from AD patients have been reported [22,23,25]. One such study from 2019 used NMR and multivariate data analysis to identify metabolites significantly correlated with dementia. This study compared diseased samples to healthy controls and samples from five years pre-diagnosis to healthy controls. The results suggested that threonine and its relevant metabolic pathways are strongly associated with dementia both pre- and post-diagnosis [22].

Another study published in 2020 identified six potential metabolites associated with Alzheimer’s disease and its preclinical stages. This study classified subjects on the AD continuum, from “cognitively normal” (CN) to Subjective Memory Decline (SMD) to Mild Cognitive Impairment (MCI) to AD. After collecting serum spectra with NMR spectroscopy, statistical analysis was performed using ROC analysis and *t*-tests. Overall, the study found that AD samples showed higher concentrations of glutamine and lower concentrations of valine compared to the CN controls [23]. This result differs slightly from the present study, which found decreased levels of both glutamine and valine amongst AD subjects. Another cohort study from 2018 considered 228 metabolites, lipids, and lipoproteins and identified ten metabolites and lipoprotein lipids with significant association to incident dementia. Lower levels of valine were correlated with an increased risk of AD, as also suggested by our results. This study also had a relatively large number of subjects (*n* = 22,623), which demonstrates its potential generalizability.

In the current pilot study, by using HRMAS, we have demonstrated a number of advancements related to blood plasma analysis for AD. Firstly, high-resolution NMR spectra can be measured with very small amounts of blood plasmas (10 μL), without any need for sample pre-treatment and with the possibility of blood metabolite quantification. Secondly, through measurements of pooled blood plasma samples, we have validated that the sample averages (i.e., pooled samples) are equivalent to the average of individual spectra, which verified the robustness of our analytic HRMAS NMR platform. Finally, blood plasma metabolomics measured with HRMAS NMR and analyzed with PCA statistical procedures have suggested the possibility of differentiating AD from non-AD groups.

In reporting our NMR-based blood plasma metabolomics, we elected to analyze data according to spectral regions, since each metabolite may contribute to multiple spectral regions. Nevertheless, we were able to identify the potential major contributing metabolites for each analyzed spectral region, making it possible to evaluate our results in association with different metabolic pathways. Since blood functions to deliver nutrients for and remove products of biological reactions, it is important to recognize that varying levels of metabolites in blood can result from simultaneous, and sometimes opposing, biological processes. For instance, an observed increase in a specific metabolite could indicate its production by a pathological process but may also reflect the over-production of this compound as a physiological response to suppress the disease. However, the complexity of explaining metabolic mechanisms does not prevent their practical utility for clinical evaluations.

Our observed metabolomics profile is capable of differentiating AD from non-AD groups, indicated by associations with several metabolic pathways, as presented in Figure 4 and Appendix A. Increased abundance of sugar phosphates and derivatives, methylated amino acids, and choline derivatives and lower abundance of organic acids appear to be associated with AD. While some of these current observations corroborate existing literature data, others might inspire further investigations. For instance, similar to our current observations, it has been shown that the blood serum levels of glucose-6-phosphate-associated glucose-6-phosphate dehydrogenase (G6PD) were found to be significantly higher in AD patients when compared with age-compatible control subjects in both sexes [7]. On the other hand, higher concentrations of the organic acid citrate in blood plasma were seen in AD patients, as opposed to the lower levels observed in our study [1].

Our observations of the increased abundance of glycine and serine in AD patients were in general agreement with published results [4,9,10,13], but existing reports considered lower blood concentrations of tryptophan and taurine in AD than in healthy controls [1,9,14]. Both more and less abundance of alanine levels in the blood have been reported from MS [10,13] and NMR [19], respectively. Treatments of anserine/carnosine supplementation (ACS) measured from peripheral blood mononuclear cells have shown suppressions of inflammatory chemokine levels and are considered to preserve verbal episodic memory, especially in elderly populations [8]. In accordance with our observations, reported data has also shown a significant reduction of carnosine levels in AD plasma compared to healthy controls [2].

Glutamine, an important glutamatergic metabolite, and its concentrations in the posterior cingulate cortex of the brain, have shown a positive correlation with its concentrations in plasma when measured with in vivo magnetic resonance spectroscopy (MRS) [11]. Large-scale genome-wide association study (GWAS) data have revealed a protective effect of circulating glutamine against AD, suggesting that more circulating glutamine might result in more available substrates during times of stress and act as a neuroprotectant [12]. However, in contrast to our observations of the decreased abundance of glutamine in AD, an HPLC study reported serum glutamine concentration to be significantly increased in AD patients [4]. This suggests that serum glutamate and glutamine levels in AD patients could vary across disease stages, potentially reflecting the progressive alteration of glutamatergic signaling during neurodegenerative processes. Of special note, a reported human plasma NMR study associated lower general cognitive ability with glutamine and ornithine [18].

In addition to the potential confounding effects resulting from variations in the measured patient populations, the apparent conflicting results may also be due to different analytic platforms. For instance, in our current study, we measured raw human plasma samples of 10 μL without any pre-treatment. As such, our measurements of the whole plasma produced very different results from those obtained from fractions of plasma after the sample-extraction procedures used for the mass spectrometry analyses cited above.

We attempted to test relationships between the blood plasma metabolomics data and cerebrospinal fluid (CSF) measurements from patient charts (Appendix A); however, no significant correlation was discovered. This is likely due to the limited sample size included in the current study and the unknown correlations between CSF and plasma NMR metabolomics, which clearly indicate an important direction for future studies.

The potential of HRMAS NMR-based human plasma metabolomics in AD characterizations revealed by this preliminary study prompts further studies in multiple directions. First, verifications with larger patient cohorts according to different clinical parameters, including age, gender, neurological status, etc., will be necessary. Due to our limited patient populations, we could only group and analyze our studied populations in a binary fashion of AD vs. non-AD. Furthermore, AD metabolomics measured from blood will need to be cross-evaluated with those measured from different origins, such as those quantified from brain tissues [17], cerebrospinal fluid [1,2,3,4], urine [2,26], saliva [27], etc. Metabolomics measured using HRMAS NMR methodology should also be compared with those evaluated from other techniques, including mass spectrometry [16] and in vivo MRI/MRS [11], to construct complete AD characterization metabolomics.

## 4. Materials and Methods

### 4.1. Patient Populations

The study was approved by the IRB of Massachusetts General Hospital. Frozen blood EDTA plasma samples of 20 μL from 35 AD patients (*n* = 16; Female = 7, Age 77.4 ± 3.6; Male = 9, Age = 78.6 ± 8.5) and non-AD subjects (*n* = 19; Female = 10, Age = 60.2 ± 7.7; Male = 9, Age = 63.2 ± 12.6) were obtained from the Massachusetts Alzheimer’s Disease Research Center (MADRC) longitudinal cohort study of cognitive aging (*n* = 30, 16 AD and 14 non-AD) and the Life Spectrum Across Aging and Neurocognition (LifeSPAN, *n* = 5 non-AD) biospecimen repositories. Detailed patient information can be found in the Appendix A.

### 4.2. Quality Assurance and Quality Control (QA/QC)

To evaluate and test the experimental accuracy, six pooled QA/QC samples were created, on ice, from the 30 MADRC plasma samples. To make these pooled controls, after thawing on ice, two well-vortexed 2.5 μL samples were extracted from each of the 30 plasma samples and mixed with 7.5 μL of D_2_O to produce controls with H_2_O:D_2_O = 10:1; then, each of the produced samples were well-vortexed and divided into three to form the final six pooled controls. All samples and controls were kept on ice whenever possible. After creating the pooled samples, all individual and pooled samples were again frozen and kept at −80 °C until the NMR analysis.

### 4.3. HRMAS NMR

All NMR measurements were conducted on a Bruker AVANCE III HD 600 MHz spectrometer (Bruker BioSpin, Billerica, MA, USA). Before measurements, all samples were allowed to thaw for approximately an hour on ice. After being vortexed, 10 μL of the individual plasma sample and 2.5 μL of D_2_O, or 12 μL of the pooled controls, were added to the 4 mm rotor, with a 12 μL Kel-f insert. HRMAS NMR data were collected at 4 °C, with a spinning rate of 3600 Hz and with a rotor synchronized CPMG method. The other spectral conditions included: recycle time 5 s, 100 CPMG *p* pulses with a total mixing time of 55.56 ms, 16 K data points with a total acquisition time of 0.85 s, and a spectral width of 16 ppm.

### 4.4. Data Analyses

Spectra analysis was conducted off-line with Bruker Topspin 3.6.2 (Bruker BioSpin, Billerica, MA, USA). The spectral processing procedure includes: 0.5 Hz line-broadening, one time zero-fill to 32 K data points, Fourier transformation, automatic and manual phasing, baseline correction, chemical shift calibration according to the up-field peak of lactate doublets at 1.32 ppm, and resonance peak curve-fitting for complete deconvolutions. Within the analyzed 5.5 to 0.5 ppm spectral region, regions between 5.1 and 4.5 ppm, and four regions of 3.68–3.55, 3.29–3.04, 2.72–2.67, and 2.59–2.52 were excluded from the data analyses due to contaminations from water and EDTA resonances. After these spectral exclusions, the total spectral intensity calculated from the rest of the deconvoluted spectral regions within 5.5–0.5 ppm, valid for plasma evaluations, was used to normalize the measured and deconvoluted spectral peak intensities. From these valid deconvoluted peaks, using 80% of all measured individual plasma samples as the threshold, 36 spectral regions were identified for further statistical analyses. Details of these regions and their potential major contributing metabolites are presented in Appendix A.

Statistical analyses on these identified regions were carried out on JMP from SAS Institute (Cary, NC, USA), including univariate analysis according to Student’s t-test (for normally distributed and equal variance data), Welch test (for normally distributed and unequal variance data), and Wilcoxon/Kruskal–Wallis test (KWW, for non-normally distributed data), as well as unsupervised multivariate principal component analyses (PCA), and receiver operating characteristic (ROC) curve analysis.

## 5. Conclusions

Since the disease’s discovery in 1906, AD diagnoses have relied on clinical evaluations of cognition rather than evaluations of the disease itself. While techniques for AD diagnosis and characterization currently remain limited, HRMAS NMR-based metabolomics may offer a new methodology. From the analysis of patient serum samples, our study found several metabolites to be significantly correlated with AD compared to controls. This study has important clinical implications as it suggests the potential of AD diagnostics from a simple, non-invasive blood sample.

## Figures and Tables

**Figure 1 metabolites-12-00181-f001:**
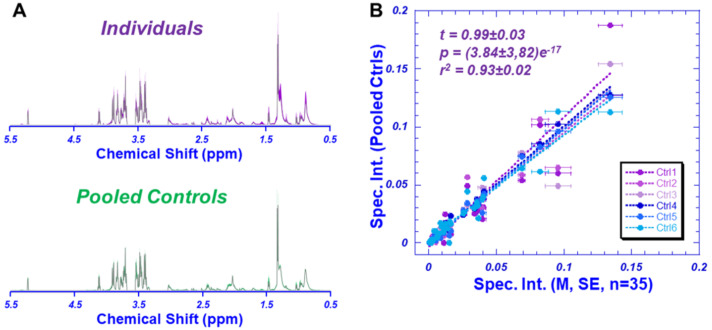
HRMAS NMR of pooled plasma control samples. (**A**) A comparison between the averaged spectrum calculated from all 35 individual samples, and the averaged spectrum calculated from the six pooled controls. The color shaded areas in the spectra represented standard deviations measured from 35 individual and six pooled samples, respectively. (**B**) Linear correlations between spectral regions between each pooled control sample and the mean of individual samples, with standard errors presented as error bars, indicating sufficient systematic QA/QC validation.

**Figure 2 metabolites-12-00181-f002:**
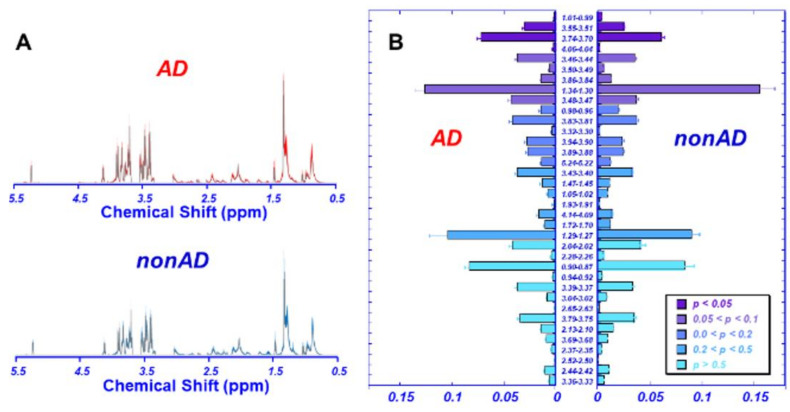
HRMAS NMR spectra measured from AD and non-AD plasma samples. (**A**) A comparison between the averaged spectra calculated from 16 AD, and 19 non-AD samples. The color shaded areas in the spectra represent standard deviations measured from each group. (**B**) Means and standard errors measured from AD and non-AD groups, respectively, are presented according to the *p* values to differentiate between the two groups.

**Figure 3 metabolites-12-00181-f003:**
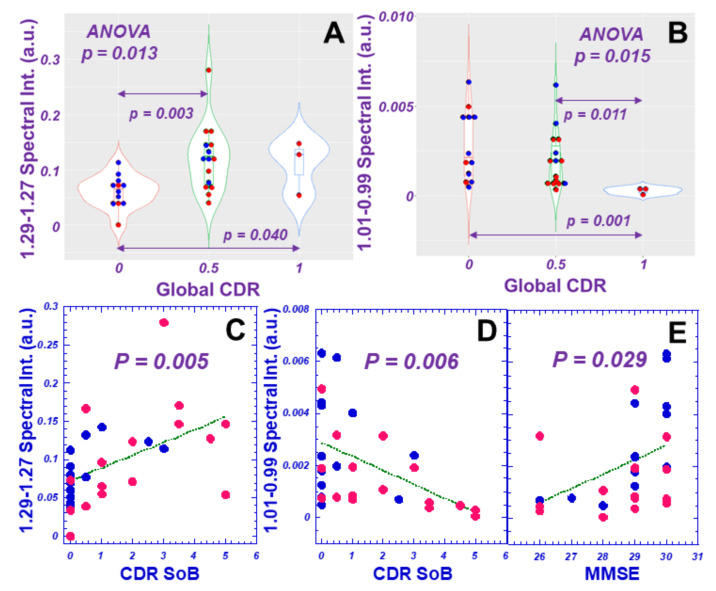
Examples of HRMAS NMR spectral regions’ demonstrated correlations with AD clinical parameters, including differentiations among Global CDR score groups (**A**,**B**), and linear relationships with CDR SoB and MMSE scores (**C**–**E**). Red dots represent AD cases, and blue dots, nonAD.

**Figure 4 metabolites-12-00181-f004:**
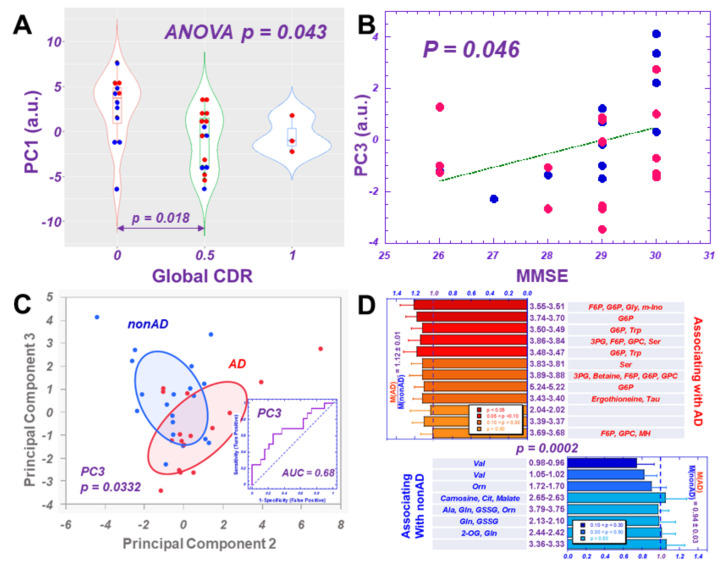
HRMAS NMR metabolomics measured from AD and non-AD plasma samples. (**A**) Principal component 1 (PC1), calculated with PCA on all 36 spectral regions, shows significant differentiations among Global CDR score groups. (**B**) PC3 linearly correlates with MMSE scores; and (**C**) presents differentiation between AD and non-AD groups; the bivariate normal ellipses represent 50%. The ROC curve calculated for PC3 indicates its capability of differentiating AD from non-AD with a 68% accuracy (insert in 4C). (**D**) Major contributing spectral regions towards PC1 and PC3 and potential unidirectionally associated metabolites with AD and non-AD differentiation, obtained from metabolomics profiles of PC1 and PC3 are presented according to the order of *p* values of univariate comparisons between AD and non-AD for each analyzed spectral region. The ratios of means of AD-associated regions (red) over those from non-AD regions (blue) are statistically different (*p* = 0.0002) for the two groups of significant associated spectral regions.

**Figure 5 metabolites-12-00181-f005:**
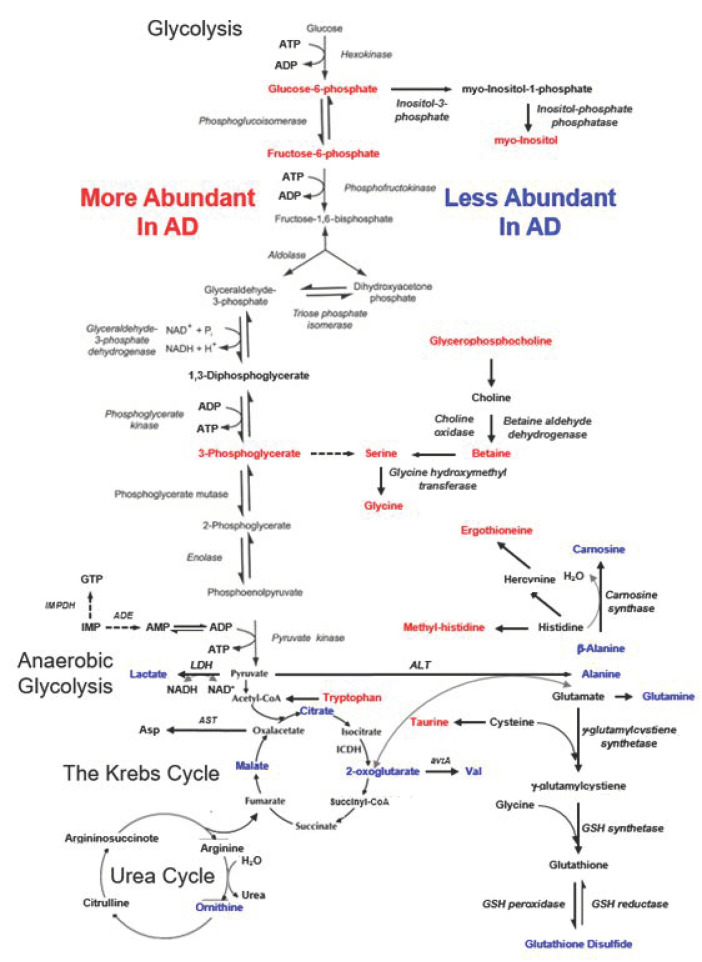
Metabolic pathways potentially altered in blood when comparing AD with non-AD cases. Red letters identify metabolites more abundant in AD, while blue letters emphasize metabolites more abundant in non-AD, or less abundant in AD, cases.

## Data Availability

Upon publication, data can be found at the MGH A. A. Martinos Center for Biomedical Imaging website.

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
