# Peer review of "A Nuclear Magnetic Resonance Spectroscopy Method in Characterization of Blood Metabolomics for Alzheimer’s Disease"

_metabolites, 2022, doi:10.3390/metabo12020181_

Round 1
Reviewer 1 Report
The manuscript A Nuclear Magnetic Resonance Spectroscopy Method in Characterization of Blood Metabolomics for Alzheimer’s Disease by Jianxiang Weng, Isabella H. Muti, Anya B. Zhong, Pia Kivisäkk, Bradley T. Hyman, Steven E. Arnold and Leo L. Cheng presents metabolomic data measured with high resolution magic angle spinning (HRMAS) nuclear magnetic resonance (NMR) on human blood plasma samples obtained from AD and non-AD subjects.
The approach used by the authors to investigate the metabolomic profile of AD and non AD subjects is High-resolution magic angle spinning (HRMAS) nuclear magnetic resonance spectroscopy (1H MRS), which is a MRS technique introduced in 1996 to enhance the resolution of MRS for non-liquid biological samples [reviewed in Kaebisch, doi:10.1002/nbm.3700]. In the present manuscript authors propose to apply this technique to human plasma in order to obtain metabolomic information and profiling from a small amount of sample.
The central statement of the manuscript appears to be the feasibility of using HRMAS to obtain metabolomic data from human plasma samples, since authors report in their discussion to have been able to detect similar results as previously shown with other metabolomic approaches.
Authors claim to have produced “high-resolution NMR spectra from blood samples with a volume of 1/5 of a drop of blood, which is commonly considered to be 50 ml” (rows 63-64). The author write that they use 10 ml of plasma sample, which is a huge amount and a blood draw of 50 ml is beyond the amount acceptable for routine examinations, being this a vial containing 5 ml of blood generating around 2 ml of plasma volume. I think that there must be an editing error in the indication of ml and that the authors probably mean ul instead. If this is the case, please correct in the manuscript. Everything would make much more sense.
In general, apart from the unusual, albeit innovative, metabolomic approach the paper is quite essential. Authors do not provide in the results section any correlations with their metabolomic results and Alzheimer’s disease stage, such as Mild cognitive impairment, early Alzheimer's Disease, nor with Braak staging [Braak, doi:10.1007/BF00308809] in case of deceased patients, for whom pathological data are available. The dataset used for producing results presented in the manuscript seems to be a retrospective one, as written in the methods section, and it may well be that more clinical data or diagnostic data, such as PET-PiB/FDG, Abeta and tau levels in the CSF or in the plasma of the patients are available. Correlations of metabolomic data also with these diagnostic results would be useful to make sense of the metabolomic data obtained, apart from the metodological statement.
Author Response
We appreciate that the reviewer caught an error in our manuscript (on page 1, line 70) caused by font conversion. We apologize for the oversight during our proofreading and have corrected 50ml to 50ul as it should be.
Based on the reviewer’s suggestion to include more clinical information on the study cohort, we have amended our supplemental table with patient demographic and clinical data (Table S1). We also have included new results obtained from metabolomics analyses against these newly included clinical parameters in a new figure (Fig 3, page 3) and revised Fig 4 (page 4). The new figure shows correlations between spectral region data and AD clinical and diagnostic parameters, as described in the complementary paragraph and caption (lines 108-118). We also showed additional correlations in Figure 4 (lines 142-153). We thank the reviewer for the valuable suggestion. However, the other parameters mentioned by the reviewer, such as Braak stages and PET data were not available with these cases.
Reviewer 2 Report
This paper is very hard to evaluate due to the many errors presented throughout with hard to understand or lose statements. The authors describe the volume of a drop of blood as 50 ml in the Introduction, for example, which is just one of the many false statements in the paper. Some of the sentences are long, impossible to comprehend or understand. Figure 4 describing intermediary metabolites of glycolysis with single carbon cycles, among others, with a purpose of quantitative comparisons indicated with different colors, uses two reference ranges, the AD and non-AD groups. Some metabolites are "upregulated" in either group, which makes no sense. Metabolite concentrations are more or less abundant, not upregulated, which is an over spilling from the personalized genomics baloney terminology so common these days, yet useless. Personalized medicine is also a useless dead end of medical research, which the authors use extensively.
This paper should be re-read, edited and re-written by senior faculty for clarity and scientific merits. In its current stage it is not sufficient for publication in any specialty journal or collection.
Author Response
We appreciate that the reviewer caught an error in our manuscript (on page 1, line 70) caused by font conversion. We apologize for the oversight during our proofreading and have corrected 50ml to 50ul as it should be
We would also like to thank the reviewer for their frank comments regarding the quality of our writing and results. We have attempted to revise the run-on sentences and typos throughout that may have made our paper hard to understand, and we hope that we have made sufficient edits to the writing style. One of their specific comments referred to the description of Figure 4 (now Figure 5), which we have edited accordingly: instead of using “under-expressed” or “over-expressed”, we now refer to the metabolites as “less abundant” or “more abundant,” per the reviewer’s recommendation (lines 155-169). Finally, with regards to this reviewer’s comment on “personalized medicine”, while this may be debated scientifically, we searched but failed to identify the use of this term in our manuscript. However, in the report, we do suggest the potential of using NMR analysis of blood samples to improve diagnosis and refine treatments for AD based on stage-specific metabolic markers. We attempted to remove any language that may cause misunderstanding and to clarify the goals of the study.
Reviewer 3 Report
The manuscript “A nuclear resonance spectroscopy method in characterization of blood metabolomics for Alzheimer’s disease” submitted by Cheng and coworkers provides the introduction of a novel NMR technique (HRMAS) for the differentiation of Alzheimer’s from healthy probands on the basis of metabolomic analysis of plasma samples. Using pooled samples the authors succeeded in the demonstration of linear correlations between each pooled sample and the mean of individual samples, and sufficient systematic quality assurance vs. quality validation.
The aim of the study has been clearly delineated, the methods used are described in sufficient detail, the studies have been properly designed, and the results have been adequately, yet concisely presented (Results) and discussed (Discussion). The manuscript may be of potential interest for both basic and clinical researchers of the area of AD and furthermore other common complex diseases, such as diabetes.
The following points should be addressed by the authors:
- The differences found between AD and healthy samples by HRMAS have to be confirmed by using independent analytical/enzymic measurements, at least for certain selected metabolites, such as glucose-6-phosphate, betaine, citrate, carnosine, glycine, glutamine. The authors mention the identity of some of the metabolites in the Discussion and Supplementary Table S3. However, this information is of qualitative nature and presented for the pooled samples, only. Quantitative data, preferably for individual samples, are required and should be included into the main text, including the methods used for their generation.
- It would be interesting to compare HRMAS NMR-based human plasma metabolomics between raw (without pretreatment) plasma samples (as exclusively performed in the present study) and fractionated samples to see the effect of sample preparation usually done for MS analyses.
Author Response
The reviewer’s comment on independent analytical/enzymic measurements are excellent and logical extensions of our current work. In fact, we indeed to consider further studies from the guidance of our metabolomics observations. However, the very reason that made this study possible was the fact that we would only need 10ul of blood plasma from these patients for the analysis. Therefore, any further independent measurements will have to be conducted when samples of large volumes are available. As for the quantitative data, such as those in Table S3 as mentioned by the reviewer, they are presented in the main text in the previous Figure 3C, or the new Figure 4D.
We appreciate the reviewer’s insight and comment. With regards to the comparison of HRMAS NMR-based human plasma metabolomics between raw (without pretreatment) plasma samples (as exclusively performed in the present study) and fractionated samples, the limitations of the current study on the available sample volumes prevented us from these analyses. However, such systematic studies are currently underway in our laboratory.
Reviewer 4 Report
A very good and thorough paper. Very well written, interesting due to the novelty of the approach. Very thorough statistics. In the introduction, maybe a better/larger technical description of the physics of the magic angle spinning compared with regular NMR would be welcome. Only research chemists and NM physicists might be familiar with these notions. As a nuclear medicine specialist, quite familiar with NMR imaging, the differential approach was a novelty for me.
Author Response
In accordance with the reviewer’s suggestion to include “a better/larger technical description of the physics of the magic angle spinning compared with regular NMR,” we added details in our introduction section to explain more of the HRMAS methodology (lines 61-67), with a newly published monographic article on the subject.
Round 2
Reviewer 1 Report
Dear Authors,
The manuscript has significantly improved and I am happy to have contribute to this. Concerning the I have noticed in the S table1 that authors have included interesting measurements regarding CSF phosphoTau181 and tTau and Abeta42/40 ratio.
Concerning this:
- I am not sure I could find the legend to S table1 in the manuscript
- Given the important CSF data available to the authors, it would be important to try to identify associations between the metabolic results with these biomarkers. I am not sure authors have shown this?
This point is quite important given the invasiveness of the CSF puncture procedure for patients. So if the metabolomic analysis performed on a small amount of easily accessible biofluid could provide a strong correlation with the CSF biomarkers, this could be a very important piece of information to provide.
Reviewer 3 Report
The authors have adequately addressed the points raised.
Author Response
We would like to thank the reviewer for their comments, and we are glad to know that we have sufficiently addressed the original edits suggested by the reviewer.